

# Association between circulating CTRP9 levels and coronary artery disease: a systematic review and meta-analysis

Ziyi Zhu[1], Qingsheng Niu[1,2,3], Shiyuan Tang[2,3] and Yaowen Jiang[2,3]

[1] West China School of Medicine, Sichuan University, Chengdu, Sichuan, China
[2] Department of Emergency Medicine, West China School of Medicine, Sichuan University, Chengdu, Sichuan, China
[3] Laboratory of Emergency Medicine, Disaster Medical Center, West China Hospital of Sichuan University, Chengdu, China

Corresponding author
Yaowen Jiang,
jywfx45613739@163.com

## ABSTRACT

**Background:** C1q tumor necrosis factor (TNF) related proteins 9 (CTRP9) is a novel adipocytokine that has been shown to have a cardioprotective effect in coronary artery disease (CAD). However, there are conflicting results on circulating levels of CTRP9 in patients with and without CAD. This meta-analysis was conducted to investigate the association between circulating CTRP9 levels and CAD.
**Objective:** The aim of this meta-analysis was to re-examine the relationship between circulating CTRP9 levels and CAD.
**Methods:** We searched PubMed, Web of Science, Embase, Cochrane Library, CNKI, VIP, Wanfang Data, and CBM for relevant studies up to October 2023, and 193 articles were identified. After reading the title, abstract and full text, a total of 25 articles were included in this meta-analysis. A prespecified protocol registered at INPLASY was followed (INPLASY202450066). Due to the high heterogeneity, we performed subgroup analyses and meta-regression based on patient characteristics, complications, clinical biochemical indicators, coronary artery lesion, and CAD classification. Publication bias was assessed using Egger's linear regression tests, Begg's rank correlation tests, and funnel plots.
**Results:** The results showed that the patient with CAD had significantly lower circulating CTRP9 levels than the control group ($Z = 3.26$, $P = 0.001$). Subgroup analysis and meta-regression findings demonstrated that observed heterogeneity could be attributed to population distribution. Patient characteristics (year of publication, patients' age, and BMI), complications (diabetes and type 2 diabetes mellitus (T2DM)), clinical biochemical indicators, coronary artery lesion (stability of coronary atherosclerotic plaque, and the number of diseased coronary vessels), and classification of CAD were not identified as source of heterogeneity.
**Conclusions:** The meta-analysis confirmed that circulating CTRP9 levels in CAD patients are significantly lower than those in patients without CAD. The association may be modified by the population distribution.

## INTRODUCTION

Coronary artery disease (CAD) is a prevalent condition among elderly individuals in China. It is a chronic inflammatory response disease of arterial intima initiated by lipid entry. During the early stages of atherosclerosis, monocytes aggregate, adhere, and migrate to the subendothelial layer of the intima where they differentiate into macrophages and finally foam cells. Subsequently, atherosclerotic plaques are formed (*Libby, 2021*). Biomarkers play a critical role in definition, prognostication and decision-making regarding cardiovascular events management (*Wang et al., 2017*). Currently, the commonly used biomarkers for CAD include cardiac troponin (cTn), creatine kinase MB (CK-MB), brain natriuretic peptide (BNP), C-reactive protein (CRP), and *etc.* (*Keffer, 1996*). However, their prognostic value is limited when assessing future development of cardiovascular disease (*Lobbes et al., 2010*). Therefore, there is a need for new biomarkers that are more sensitive and accurate for early diagnosis of CAD.

C1q tumor necrosis factor (TNF) related proteins (CTRPs), a highly conserved family of adiponectin paralogs, is a well-known homeostatic factor that regulate glucose levels, lipid metabolism, and insulin sensitivity through its anti-inflammatory, anti-fibrotic, and antioxidant effects. Researches indicated that CTRP1 and CTRP5 may serve as potential risk factors for CAD, whereas CTRP3, CTRP9, CTRP12, and CTRP13 function as protective factors (*Si, Fan & Sun, 2020*). Among all the CTRP paralogs, CTRP9 exhibits the highest degree of amino acid identity to adiponectin, which possesses anti-atherogenic properties, in its globular C1q domain (*Liu et al., 2022a*; *Shanaki et al., 2020*; *Wong et al., 2009*). The expression of CTRP9 in adipocytes is significantly lower than adiponectin, but far exceeds in cardiac tissue (*Peterson, Wei & Wong, 2009*). This discovery has heightened awareness regarding the pivotal role of CTRP9 in cardiovascular disease.

CTRP9 is mainly expressed in adipose tissue and interstitial vascular cells (*Schäffler & Buechler, 2012*). *Sun et al. (2013)* have found a significant reduction in the average expression of CTRP9 in both adipocytes and plasma of mice after myocardial infarction. *In vivo* CTRP9 administration to mice improved their condition by enhancing survival rate and systolic function recovery. Moreover, overexpression of CTRP9 has been shown to alleviate myocardial ischemia-reperfusion injury (IRI) and improve cardiac function (*Kambara et al., 2012*). It has also been demonstrated that CTRP9 can inhibit superoxide production in diabetes mice (*Su et al., 2013*).

Numerous reports have indicated that low CTRP9 levels can serve as an independent risk factor for CAD. However, the existing studies primarily concentrate on animal models, with a dearth of studies examining the role of CTRP9 in CAD occurrence (*Li et al., 2013*). Recently, conflicting results regarding the association between circulating levels of CTRP9 and CAD have emerged. Therefore, our study aims to investigate association between circulating levels of CTRP9 and CAD.

## MATERIALS AND METHODS

### Search strategy

The following databases were searched for relevant studies by two authors (Ziyi Zhu and Qingsheng Niu): PubMed, Web of Science, Embase, Cochrane Library, CNKI, VIP, Wan

fang Data, and CBM. The retrieval time spanned from the inception of each database to October 2023. Language restrictions were not applied during the search process and references were traced to avoid omissions. A prespecified protocol registered at INPLASY was followed (INPLASY202450066), and portions of this text were previously published as part of a preprint (*Zhu et al., 2024*). Our search strategy was as follows: (CTRP9 OR C1q TNF related protein 9 OR C1q Tumor Necrosis Factor Related Protein 9) AND (Coronary Heart Disease OR Coronary Artery Disease OR Coronary Atherosclerotic Heart Disease). This process iterated until no further relevant article was identified. If there was any disagreement, a third researcher (Shiyuan Tang) was included to discuss and establish consensus.

## Inclusion and exclusion criteria

Titles and abstracts were initially screened to determine if they met the inclusion criteria for further evaluation. Subsequently, full-text articles were assessed based on these criteria as well.

The following inclusion criteria were used: (1) studies involving adults with or without complications who developed CAD; (2) CAD serving as the exposure factor with patients meeting diagnostic criteria included; (3) comparison groups consisting of adults without CAD during the same period; (4) assessment of circulating levels of CTRP9 as an outcome measure; (5) cohort studies and random controlled studies. Exclusion criteria consisted of: (1) articles lacking valid data; (2) duplicate publications; (3) withdrawn articles.

## Data collection and quality assessment

We extracted the following data from the included articles: number of patients and controls, first author, year of publication, study area, age, body mass index (BMI), relevant clinical biochemical indicators, complications, and circulating CTRP9 levels of patients and controls. The risk and bias of all selected studies were evaluated using the Newcastle Ottawa Scale (NOS) (*Wells et al., 2014*). Each study can earn up to nine points. If a score of 0–3 was obtained, it is considered a low-quality article. Scoring 4–6 points was considered medium quality, and 7–9 was considered high-quality. Two autonomous researchers (Ziyi Zhu and Qingsheng Niu) conducted data extraction and assessed the quality of references, followed by a cross-validation process. A third researcher (Shiyuan Tang) was included to conduct the discussion and reach a consensus if there was any disagreement.

## Statistical analysis

We used mean and standard deviation (SD) to describe the data extracted from the studies, and standardized mean differences (SMDs) with 95% CI were chosen to express the continuous variable data. We employed the Q-test and $I^2$ statistic for heterogeneity testing. An $I^2$ value of less than 50% indicated no statistical heterogeneity, in which case a fixed effects model was applied for calculation. Conversely, if there was significant heterogeneity, a random effects model would be utilized. The publication bias was assessed using Egger's linear regression tests, Begg's rank correlation tests, and funnel plot. Additionally, we

conducted leave-1-out sensitivity analysis by excluding each study one by one to observe any alteration of the direction of SMDs. A *p*-value less than 0.05 was considered statistically significant in our analysis. Statistical analysis was performed using Review Manager 5.3 and STATA 15.1.

# RESULTS

## Search

A total of 193 articles were obtained through initial screening. After reviewing the titles and abstracts, 118 articles were excluded. Among the 118 excluded articles, there were 17 that lacked a non CAD control group, five unrelated to CAD, 28 on other molecules in CTRPs family and 65 on the genes or pathogenesis of CTRP9. Additionally, three articles were withdrawn. After removing duplicates, twenty-nine articles were left. We carefully examined the remaining 29 articles while eliminating those with repeated data sources or unavailable full texts. Ultimately, we included a total of 25 studies that met our inclusion and exclusion criteria (Fig. 1).

## Study characteristics

Twenty-five studies were included in this meta-analysis. Among them, five articles investigated levels of CTRP9 in patients with both CAD and diabetes concurrently. Four articles assessed the levels of CTRP9 in patient groups with varying stability of coronary plaques. Additionally, four articles categorized CAD into three types: stable angina pectoris (SAP), unstable angina pectoris (UAP), and acute myocardial infarction (AMI). Six articles focused on patient groups with different extent of the coronary atherosclerosis lesions (Table 1).

## Quality assessment

The quality assessment of included articles was conducted using the NOS scale (Table 2). Among the included articles, twenty-two were deemed to be of high quality (88%, 22/25), while three were classified as medium quality (12%, 3/25), and no article was rated as low quality.

## Circulating levels of CTRP9

The results of the meta-analysis revealed a significant decrease in circulating CTRP9 levels among patients with CAD compared to the control group (Z = 3.26, *P* = 0.001) (Fig. 2). Considering the high heterogeneity, statistical analysis was conducted using a random effects model. Leave-1-out sensitivity analysis was used to assess the impact of each individual study on the overall results. Notably, consistent findings were observed before and after excluding each study one by one.

## Subgroup analysis

Given its high heterogeneity, we analyzed subgroups based on the following five aspects: (1) the main characteristics of articles (year of publication, study area, patients' age, and BMI); (2) complications (diabetes or type 2 diabetes mellitus (T2DM)); (3) relevant clinical

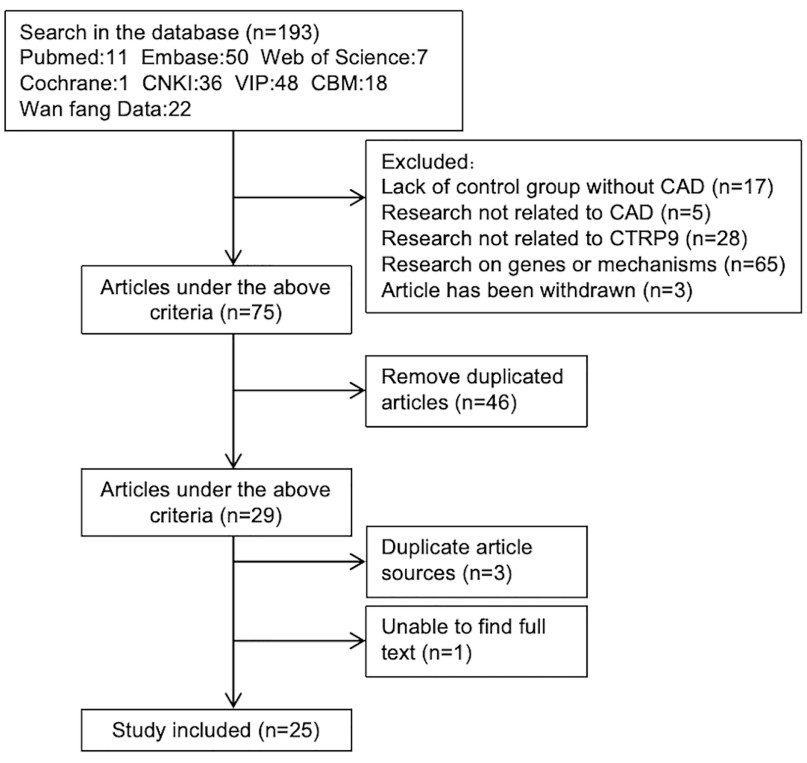

**Figure 1  Articles screening process.**     

biochemical indicators (triglyceride (TG), low-density lipoprotein (LDL), and high-density lipoprotein (HDL)); (4) coronary artery lesion (stability of coronary atherosclerotic plaque and the number of diseased coronary vessels); (5) classification of CAD (UAP, SAP, AMI). Meta-regression was used to assess the between-subgroup heterogeneity (Table 3).

### Characteristics of articles
The main characteristics of the articles we analyzed included year of publication, study area, patients' BMI, and age. Subgroup analysis was conducted based on these features. The included articles were published between 2013 and 2023, with 2017 serving as the boundary. Articles published before 2017 constituted a subgroup, while those published after 2017 formed another. The result revealed insignificant between-subgroup heterogeneity ($P = 0.483$), indicating that the year of publication was not the source of heterogeneity. In addition, CTRP9 was an adipocytokine that is homologous to adiponectin, so CTRP9 levels might be related to obesity. According to international BMI classification standards, individuals with a BMI < 25 were considered within the normal range, while those with a BMI between 25 and 30 were classified as overweight, and a BMI > 30 indicated obesity. Subgroup analysis was performed based on the criteria mentioned above, showing insignificant heterogeneity among subgroups ($P = 0.517$). Regarding patients' age, the included studies focused on individuals aged 50 to 70 years old. Therefore, we divided the studies into four subgroups for analysis. The results suggested no statistical significance in heterogeneity ($P = 0.904$) among the four subgroups: patients aged 50–55, 55–60, 60–65, and >65 years old. Consequently, it can be concluded that

**Table 1 Basic characteristics of articles.**

| First author | Year of publication | n CG/PG | Age | | CTRP9 mean | | CTRP9 SD | |
|---|---|---|---|---|---|---|---|---|
| | | | CG | PG | CG | PG | CG | PG |
| 1. *Wang & Li (2016)* | 2016 | 40/68 | 55.26 | 56.7 | 176.31 | – | 21.57 | – |
| *CHD (a)* | | 40/36 | | | | 116.73 | | 10.36 |
| *CHD+T2DM (b)* | | 40/32 | | | | 89.91 | | 9.01 |
| 2. *Chen, Lin & Huang (2022)* | 2022 | 74/124 | 63.34 | 72.36 | 41.83 | 35.92 | 3.72 | 2.43 |
| 3. *Shi & Cui (2018)* | 2018 | 60/146 | 62.1 | 62.8 | 3.79 | 3.44 | 0.26 | 0.3 |
| *Soft plaques* | | 60/50 | | | | 3.01 | | 0.24 |
| *Mixed plaques* | | 60/49 | | | | 3.6 | | 0.3 |
| *Hard plaques* | | 60/47 | | | | 3.51 | | 0.27 |
| 4. *Cheng (2019)* | **2019** | 56/56 | 65.09 | 64.32 | 3.79 | 3.23 | 0.21 | 0.32 |
| *Soft plaques* | | 56/17 | | | | 2.01 | | 0.19 |
| *Mixed plaques* | | 56/19 | | | | 3.47 | | 0.32 |
| *Hard plaques* | | 56/20 | | | | 3.66 | | 0.23 |
| 5. *Yu & Lu (2021)* | 2021 | 60/60 | 67.53 | 67.48 | 3.75 | 3.02 | 1.23 | 0.62 |
| *Soft plaques* | | 60/20 | | | | 3.02 | | 0.25 |
| *Mixed plaques* | | 60/21 | | | | 3.52 | | 0.65 |
| *Hard plaques* | | 60/19 | | | | 3.66 | | 0.68 |
| 6. [1] *Wang (2014b)* | 2014 | 21/67 | 62.286 | 62.507 | 101.031 | 78.919 | 31.87 | 35.57 |
| 7. [2] *Wang et al. (2014)* | 2014 | 21/67 | 62.5 | | 91.31 | – | 68.43 | – |
| *CHD (a)* | | 21/39 | | | | 50.28 | | 55.21 |
| *CHD+diabetes (b)* | | 21/28 | | | | 22.21 | | 23.07 |
| 8. *Chen et al. (2019)* | 2019 | 28/86 | 58.54 | 59.47 | 1.98 | 1.19 | 0.71 | 0.54 |
| *Single vessel lesion* | | 28/21 | | | | 1.49 | | 0.63 |
| *Double vessel lesions* | | 28/25 | | | | 1.27 | | 0.48 |
| *Multi vessel lesions* | | 28/40 | | | | 0.98 | | 0.42 |
| 9. *Shi (2015)* | 2015 | 26/60 | 61.2 | 60.1 | 3.85 | – | 0.27 | – |
| *UAP (c)* | | 26/20 | | | | 3.46 | | 0.22 |
| *AMI (d)* | | 26/20 | | | | 3.38 | | 0.18 |
| *SAP (e)* | | 26/20 | | | | 3.61 | | 0.18 |
| *Single vessel lesion (f)* | | 26/12 | | | | 3.43 | | 0.08 |
| *Double vessel lesions (g)* | | 26/20 | | | | 3.58 | | 0.13 |
| *Multi vessel lesions (h)* | | 26/28 | | | | 3.39 | | 0.23 |
| 10. *Yang & Li (2020)* | 2020 | 40/80 | **74.25** | 74.67 | 14.21 | – | 2.36 | – |
| *CHD (a)* | | 40/40 | | | | 10.21 | | 2.45 |
| *CHD+T2DM (b)* | | 40/40 | | | | 6.54 | | 3.02 |
| 11. *Liu et al. (2014)* | 2014 | 26/61 | 57.96 | 60.13 | 3.706 | 3.464 | 0.246 | 0.188 |
| *UAP* | | 26/18 | | | | 3.471 | | 0.204 |
| *AMI* | | 26/18 | | | | 3.363 | | 0.188 |
| *SAP* | | 26/25 | | | | 3.531 | | 0.147 |
| *Single vessel lesion* | | 26/12 | | | | 3.537 | | 0.082 |
| *Double vessel lesions* | | 26/21 | | | | 3.517 | | 0.128 |
| *Multi vessel lesions* | | 26/28 | | | | 3.393 | | 0.232 |

| Table 1 (continued) | | | | | | | | |
|---|---|---|---|---|---|---|---|---|
| First author | Year of publication | n CG/PG | Age | | CTRP9 mean | | CTRP9 SD | |
| | | | CG | PG | CG | PG | CG | PG |
| **12. *Shi et al. (2021)*** | 2021 | 100/180 | 57.11 | 56.38 | 39.74 | 32.83 | 2.45 | 1.93 |
| *Single vessel lesion* | | 100/84 | | | | 34.12 | | 2.38 |
| *Double vessel lesions* | | 100/61 | | | | 32.94 | | 1.92 |
| *Multi vessel lesions* | | 100/35 | | | | 29.54 | | 2.27 |
| **13. *Guo (2018)*** | 2018 | 28/78 | 55.11 | 54.7 | 134.39 | 83.67 | 22.3 | 28.3 |
| *Single vessel lesion* | | 28/27 | | | | 113.3 | | 16.65 |
| *Double vessel lesions* | | 28/31 | | | | 80.28 | | 16.75 |
| *Multi vessel lesions* | | 28/20 | | | | 51.61 | | 15.44 |
| **14. *Li et al. (2013)*** | 2013 | 113/278 | 63.81 | 65.64 | 136.36 | 120.38 | 50.04 | 35.83 |
| *Single vessel lesion* | | 113/133 | | | | 123.7 | | 39.4 |
| *Double vessel lesions* | | 113/83 | | | | 121.87 | | 35.66 |
| *Multi vessel lesions* | | 113/62 | | | | 111.25 | | 25.47 |
| **15. *Hu & Lin (2021)*** | 2021 | 44/159 | 60.41 | 63.45 | 215.203 | 119.267 | 28.497 | 20.44 |
| **16. *Ahmed et al. (2018)*** | 2018 | 13/44 | 51.15 | 56.38 | 304.46 | – | 34.61 | – |
| *CAD (a)* | | 13/29 | | | | 194.9 | | 18 |
| *CAD+T2DM (b)* | | 13/15 | | | | 101.4 | | 22.08 |
| **17. *Jiang et al. (2021)*** | 2021 | 58/79 | 63.71 | | 3.53 | 3.31 | 0.2 | 0.19 |
| **18. *Moradi et al. (2018)*** | 2018 | 80/220 | 57.03 | 58.335 | 148.7 | – | 4 | – |
| *CAD (a)* | | 80/157 | | | | 202 | | 4.9 |
| *CAD+T2DM (b)* | | 80/63 | | | | 211.2 | | 6.8 |
| **19. *Wang et al. (2015)*** | 2015 | 121/241 | 61.29 | 62.51 | 96.14 | 83.89 | 33.13 | 36.18 |
| **20. *Liu et al. (2022b)*** | 2022 | 79/210 | – | – | 135.51 | 119.117 | 46.426 | 24.335 |
| **21. *Dong et al. (2021)*** | **2021** | **78/86** | 59.44 | 61.13 | 12.15 | 9.08 | 3.24 | 2.7 |
| *Soft plaques* | | 78/21 | | | | 6.82 | | 2.5 |
| *Mixed plaques* | | 78/10 | | | | 8.78 | | 2.2 |
| *Hard plaques* | | 78/55 | | | | 10.74 | | 2.36 |
| **22. *Wang (2014a)*** | 2014 | 80/120 | 56.1 | 57.5 | 140.33 | 117.87 | 38.27 | 42.19 |
| **23. *Si (2021)*** | **2021** | **20/26** | **6**1 | 62 | 19.92 | 17.09 | 5.21 | 2.06 |
| **24. *Hu, Wang & Zhao (2023)*** | 2023 | 50/120 | 61.19 | – | 1.93 | – | 0.33 | – |
| *ACS (i)* | | 50/65 | | 61.1 | | 3.02 | | 0.45 |
| *SAP (e)* | | 50/55 | | 63.02 | | 3.54 | | 0.51 |
| **25. *Liu et al. (2016)*** | 2016 | 30/32 | 67.1 | 68.1 | 100.82 | 43.04 | 9.98 | 4.33 |

**Note:**
CG, control group; PG, patients group; n, the number of population in control and patients group; [1] and [2], two articles conducted by one author; a, CAD patients without complications; b, CAD patients with complications; c, patients with UAP; d, patients with AMI; e, patients with SAP; f, patients with single coronary vessel lesion; g, patients with double coronary vessel lesions; h, patients with multi coronary vessel lesions; i, patients with acute coronary syndrome (ACS).

neither patients' BMI nor age contributed to the observed heterogeneity. In terms of population distribution, most objects in the studies were Chinese, and population distribution was a cause of heterogeneity ($P = 0.005$).

**Table 2 All primer sequences used in the experiment.**

| Study | Item | | | | | | | | Score |
|-------|------|------|------|------|------|------|------|------|-------|
| | 1 | 2 | 3 | 4 | 5 | 6 | 7 | 8 | |
| Wang & Li (2016) | 1 | 1 | 1 | 1 | 2 | 1 | 0 | 0 | 7 |
| Chen, Lin & Huang (2022) | 1 | 1 | 1 | 1 | 2 | 1 | 0 | 0 | 7 |
| Shi & Cui (2018) | 1 | 1 | 1 | 1 | 2 | 1 | 0 | 0 | 7 |
| Cheng (2019) | 1 | 1 | 1 | 1 | 0 | 1 | 0 | 0 | 5 |
| Yu & Lu (2021) | 1 | 1 | 1 | 1 | 2 | 1 | 0 | 0 | 7 |
| Wang (2014b)[a] | 1 | 1 | 1 | 1 | 2 | 1 | 0 | 0 | 7 |
| Wang et al. (2014)[b] | 1 | 1 | 1 | 1 | 2 | 1 | 0 | 0 | 7 |
| Chen et al. (2019) | 1 | 1 | 1 | 1 | 2 | 1 | 0 | 0 | 7 |
| Shi (2015) | 1 | 1 | 1 | 1 | 2 | 1 | 0 | 0 | 7 |
| Yang & Li (2020) | 1 | 1 | 1 | 1 | 2 | 1 | 0 | 0 | 7 |
| Liu et al. (2014) | 1 | 1 | 1 | 1 | 2 | 1 | 0 | 0 | 7 |
| Shi et al. (2021) | 1 | 1 | 1 | 1 | 2 | 1 | 0 | 0 | 7 |
| Guo (2018) | 1 | 1 | 1 | 1 | 2 | 1 | 0 | 0 | 7 |
| Li et al. (2013) | 1 | 1 | 1 | 1 | 2 | 1 | 0 | 0 | 7 |
| Hu & Lin (2021) | 1 | 1 | 1 | 1 | 2 | 1 | 1 | 1 | 9 |
| Dong et al. (2021) | 1 | 1 | 1 | 1 | 2 | 1 | 0 | 0 | 7 |
| Wang (2014a) | 1 | 0 | 1 | 1 | 2 | 1 | 0 | 0 | 6 |
| Si (2021) | 1 | 1 | 1 | 1 | 2 | 1 | 0 | 0 | 7 |
| Hu, Wang & Zhao (2023) | 1 | 1 | 1 | 1 | 2 | 1 | 0 | 0 | 7 |
| Liu et al. (2016) | 1 | 1 | 1 | 1 | 2 | 1 | 0 | 0 | 7 |
| Ahmed et al. (2018) | 1 | 1 | 0 | 1 | 2 | 1 | 0 | 0 | 6 |
| Jiang et al. (2021) | 1 | 1 | 1 | 1 | 2 | 1 | 1 | 0 | 8 |
| Moradi et al. (2018) | 1 | 1 | 1 | 1 | 2 | 1 | 0 | 0 | 7 |
| Wang et al. (2015) | 1 | 1 | 1 | 1 | 2 | 1 | 0 | 0 | 7 |
| Liu et al. (2022b) | 1 | 1 | 1 | 1 | 2 | 1 | 0 | 0 | 7 |

**Note:**
[a] and [b]: two articles conducted by one author.

## Complication

The overexpressing of CTRP9 using adenovirus vectors has been shown to effectively reduce blood glucose and insulin levels in ob/ob mice, suggesting a protective effect of CTRP9 against diabetes (*Wong et al., 2009*). Therefore, we conducted subgroup analysis based on the presence of diabetes or T2DM among CAD patients. The results revealed insignificant heterogeneity between subgroups, indicating that neither diabetes ($P = 0.577$) nor T2DM ($P = 0.816$) was a source of heterogeneity.

## Clinical indicator

Abnormal lipid metabolism was closely related to atherosclerosis, so we conducted subgroup analysis by categorizing clinical lipid metabolism indicators, including TG, LDL, and HDL. TG was categorized with a threshold value of 1.7 mmol/L, LDL was grouped with a threshold of 3.12 mmol/L, and HDL was categorized with a threshold value of

| Study or Subgroup | Experimental Mean | SD | Total | Control Mean | SD | Total | Weight | Std. Mean Difference IV, Random, 95% CI |
|---|---|---|---|---|---|---|---|---|
| Chen MZ 2022 | 35.92 | 2.43 | 124 | 41.83 | 3.72 | 74 | 2.9% | −1.98 [−2.33, −1.63] |
| Chen YL 2019 | 1.19 | 0.54 | 86 | 1.98 | 0.71 | 28 | 2.9% | −1.34 [−1.80, −0.88] |
| Cheng XL 2019 | 3.23 | 0.32 | 56 | 3.79 | 0.21 | 56 | 2.9% | −2.05 [−2.52, −1.59] |
| Dong QT 2021 | 9.08 | 2.7 | 86 | 12.15 | 3.24 | 78 | 2.9% | −1.03 [−1.36, −0.70] |
| Guo J 2018 | 83.67 | 28.3 | 78 | 134.39 | 22.3 | 28 | 2.8% | −1.87 [−2.38, −1.37] |
| Hu YN 2021 | 119.267 | 20.44 | 159 | 215.203 | 28.497 | 44 | 2.8% | −4.27 [−4.80, −3.73] |
| Hu ZS (e) | 3.54 | 0.51 | 55 | 1.93 | 0.33 | 50 | 2.8% | 3.68 [3.05, 4.32] |
| Hu ZS (i) | 3.02 | 0.45 | 65 | 1.93 | 0.33 | 50 | 2.8% | 2.69 [2.18, 3.20] |
| Jiang N 2021 | 3.31 | 0.19 | 79 | 3.53 | 0.2 | 58 | 2.9% | −1.13 [−1.49, −0.76] |
| Li MC 2013 | 120.38 | 35.83 | 278 | 136.36 | 50.04 | 113 | 2.9% | −0.39 [−0.61, −0.17] |
| Liu SS 2016 | 43.04 | 4.33 | 32 | 100.82 | 9.98 | 30 | 2.4% | −7.50 [−8.96, −6.05] |
| Liu TJ 2014 | 3.464 | 0.188 | 61 | 3.706 | 0.246 | 26 | 2.8% | −1.16 [−1.65, −0.67] |
| Liu YX 2022 | 119.117 | 24.335 | 210 | 135.51 | 46.426 | 79 | 2.9% | −0.51 [−0.77, −0.25] |
| Nariman Moradi (a) | 202 | 4.9 | 157 | 148.7 | 4 | 80 | 2.6% | 11.51 [10.43, 12.59] |
| Nariman Moradi (b) | 211.2 | 6.8 | 63 | 148.7 | 4 | 80 | 2.5% | 11.49 [10.10, 12.88] |
| Sara F. Ahmed (a) | 194.9 | 18 | 29 | 304.46 | 34.61 | 13 | 2.6% | −4.44 [−5.63, −3.25] |
| Sara F. Ahmed (b) | 101.4 | 22.08 | 15 | 304.46 | 34.61 | 13 | 2.1% | −6.90 [−8.99, −4.82] |
| Shi HX (c) | 3.46 | 0.22 | 20 | 3.85 | 0.27 | 26 | 2.8% | −1.54 [−2.20, −0.87] |
| Shi HX (d) | 3.38 | 0.18 | 20 | 3.85 | 0.27 | 26 | 2.8% | −1.96 [−2.68, −1.24] |
| Shi HX (e) | 3.61 | 0.18 | 20 | 3.85 | 0.27 | 26 | 2.8% | −1.00 [−1.62, −0.38] |
| Shi HX (f) | 3.43 | 0.08 | 12 | 3.85 | 0.27 | 26 | 2.7% | −1.79 [−2.60, −0.99] |
| Shi HX (g) | 3.58 | 0.13 | 20 | 3.85 | 0.27 | 26 | 2.8% | −1.20 [−1.84, −0.57] |
| Shi HX (h) | 3.39 | 0.23 | 28 | 3.85 | 0.27 | 26 | 2.8% | −1.81 [−2.45, −1.17] |
| Shi JH 2018 | 3.44 | 0.3 | 146 | 3.79 | 0.26 | 60 | 2.9% | −1.21 [−1.53, −0.88] |
| Shi XC 2021 | 32.83 | 1.93 | 180 | 39.74 | 2.45 | 100 | 2.9% | −3.24 [−3.60, −2.87] |
| Si YQ 2021 | 17.09 | 2.06 | 26 | 19.92 | 5.21 | 20 | 2.8% | −0.74 [−1.34, −0.14] |
| Wang C (a) | 116.73 | 10.36 | 36 | 176.31 | 21.57 | 40 | 2.8% | −3.43 [−4.15, −2.71] |
| Wang C (b) | 89.91 | 9.01 | 32 | 176.31 | 21.57 | 40 | 2.7% | −4.97 [−5.93, −4.02] |
| Wang J 2015 | 83.89 | 36.18 | 241 | 96.14 | 33.13 | 121 | 2.9% | −0.35 [−0.57, −0.13] |
| Wang JF 2014 | 117.87 | 42.19 | 120 | 140.33 | 38.27 | 80 | 2.9% | −0.55 [−0.84, −0.26] |
| Wang Q(1) 2014 | 78.919 | 35.57 | 67 | 101.037 | 31.87 | 21 | 2.8% | −0.63 [−1.13, −0.13] |
| Wang Q(2)(a) 2014 | 50.28 | 55.21 | 39 | 91.31 | 68.43 | 21 | 2.8% | −0.67 [−1.22, −0.13] |
| Wang Q(2)(b) 2014 | 22.21 | 23.07 | 28 | 91.31 | 68.43 | 21 | 2.8% | −1.42 [−2.06, −0.78] |
| Yang HY (a) | 10.21 | 2.45 | 40 | 11 | 2.36 | 40 | 2.9% | −0.33 [−0.77, 0.12] |
| Yang HY (b) | 6.54 | 3.02 | 40 | 11 | 2.36 | 40 | 2.8% | −1.63 [−2.14, −1.12] |
| Yu LF 2021 | 3.02 | 0.62 | 60 | 3.75 | 1.23 | 60 | 2.9% | −0.74 [−1.12, −0.37] |
| **Total (95% CI)** | | | 2808 | | | 1720 | 100.0% | −0.94 [−1.50, −0.37] |

Heterogeneity: Tau² = 2.86; Chi² = 1962.86, df = 35 (P < 0.00001); I² = 98%
Test for overall effect: Z = 3.26 (P = 0.001)



**Figure 2** Meta-analysis results of circulating CTRP9 levels in non CAD and CAD patients (*Chen, Lin & Huang, 2022*; *Chen et al., 2019*; *Cheng, 2019*; *Dong et al., 2021*; *Guo, 2018*; *Hu & Lin, 2021*; *Hu, Wang & Zhao, 2023*; *Jiang et al., 2021*; *Li et al., 2013*; *Liu et al., 2016, 2014, 2022b*; *Moradi et al., 2018*; *Ahmed et al., 2018*; *Shi, 2015*; *Shi & Cui, 2018*; *Shi et al., 2021*; *Si, 2021*; *Wang & Li, 2016*; *Wang et al., 2015*; *Wang, 2014a, 2014b, Wang et al., 2014*; *Yang & Li, 2020*; *Yu & Lu, 2021*).

**Table 3 The results of subgroup analysis.**

| Subgroup | Number of comparisons | SMD (95% CI) | Z value | P | Test of heterogeneity I²(%) | P | Pa |
|---|---|---|---|---|---|---|---|
| **Overall** | 36 | [−1.50 to −0.37] | 3.26 | 0.001 | 98 | <0.001 | |
| 1. *Year of publication* | 36 | | | | | | 0.483 |
| Before 2017 | 16 | [−2.25 to −1.23] | 6.72 | <0.001 | 95 | <0.001 | |
| 2017–2023 | 20 | [−1.31 to 0.64] | 0.67 | 0.5 | 99 | <0.001 | |
| 2. *Age* | 35 | | | | | | 0.904 |
| 50–55 | 1 | [−2.38 to −1.37] | 7.31 | <0.001 | – | – | |
| 55–60 | 9 | [−2.63 to 2.23] | 0.16 | 0.87 | 99 | <0.001 | |
| 60–65 | 19 | [−1.56 to −0.29] | 2.85 | 0.004 | 97 | <0.001 | |

(Continued)

| Table 3 (continued) | | | | | | | |
| Subgroup | Number of comparisons | SMD (95% CI) | Z value | P | Test of heterogeneity | | P^a |
| | | | | | I²(%) | P | |
| >65 | 6 | [−3.31 to −1.25] | 4.35 | <0.001 | 97 | <0.001 | |
| 3. *Population* | 36 | | | | | | 0.005 |
| Chinese | 32 | [−1.83 to −0.94] | 6.07 | <0.001 | 97 | <0.001 | |
| Non-Chinese | 4 | [−6.56 to 12.42] | 0.61 | 0.54 | 99 | <0.001 | |
| 4. *TG* | 20 | | | | | | 0.179 |
| <1.7 | 15 | [−2.69 to −1.42] | 6.31 | <0.001 | 97 | <0.001 | |
| >1.7 | 5 | [−1.66 to −0.47] | 3.50 | <0.001 | 93 | <0.001 | |
| 5. *LDL* | 18 | | | | | | 0.180 |
| <3.12 | 15 | [−1.99 to −1.01] | 6.01 | <0.001 | 96 | <0.001 | |
| >3.12 | 3 | [−5.71 to −0.20] | 2.1 | 0.04 | 98 | <0.001 | |
| 6. *HDL* | 20 | | | | | | 0.162 |
| <1.04 | 6 | [−3.69 to −1.03] | 3.47 | <0.001 | 98 | <0.001 | |
| >1.04 | 14 | [−1.95 to −1.06] | 6.69 | <0.001 | 93 | <0.001 | |
| 7. *Complication* | | | | | | | |
| (1) Diabetes | 11 | | | | | | 0.577 |
| CAD only | 5 | [−3.98 to 4.04] | 0.01 | 0.99 | 99 | <0.001 | |
| CAD+diabetes | 6 | [−6.14 to 1.73] | 1.1 | 0.27 | 99 | <0.001 | |
| (2) T2DM | 8 | | | | | | 0.816 |
| CAD only | 4 | [−5.18 to, 6.16] | 0.17 | 0.87 | 99 | <0.001 | |
| CAD+T2DM | 4 | [−7.24 to 5.67] | 0.24 | 0.81 | 99 | <0.001 | |
| 8. *Stability of coronary atherosclerotic plaque* | 12 | | | | | | 0.075 |
| Soft plaques | 4 | [−5.31 to −1.39] | 3.35 | <0.001 | 97 | <0.001 | |
| Mixed plaques | 4 | [−1.24 to −0.33] | 3.36 | <0.001 | 68 | 0.03 | |
| Hard plaques | 4 | [−0.95 to −0.19] | 2.91 | 0.004 | 67 | 0.03 | |
| 9. *Number of diseased coronary vessels* | 18 | | | | | | 0.152 |
| Single vessel lesions | 6 | [−1.95 to −0.36] | 2.84 | 0.005 | 94 | <0.001 | |
| Double vessel lesions | 6 | [−2.52 to −0.56] | 3.08 | 0.002 | 96 | <0.001 | |
| Multi vessel lesions | 6 | [−3.44 to −1.09] | 3.78 | <0.001 | 96 | <0.001 | |
| 10. *Types of CAD* | 9 | | | | | | 0.241 |
| UAP | 2 | [−1.78 to −0.74] | 4.74 | <0.001 | 21 | 0.26 | |
| AMI | 2 | [−2.22 to −1.23] | 6.81 | <0.001 | 0 | 0.36 | |
| SAP | 5 | [−1.16 to 1.29] | 0.11 | 0.91 | 98 | <0.001 | |
| 11. *BMI* | 24 | | | | | | 0.517 |
| <25 | 8 | [−1.93 to 2.75] | 0.34 | 0.73 | 99 | <0.001 | |
| 25–30 | 15 | [−1.72 to −0.22] | 2.52 | 0.01 | 97 | <0.001 | |
| >30 | 1 | [−8.99 to −4.82] | 6.48 | <0.001 | – | – | |

**Note:**
P^a, the results of meta-regression.

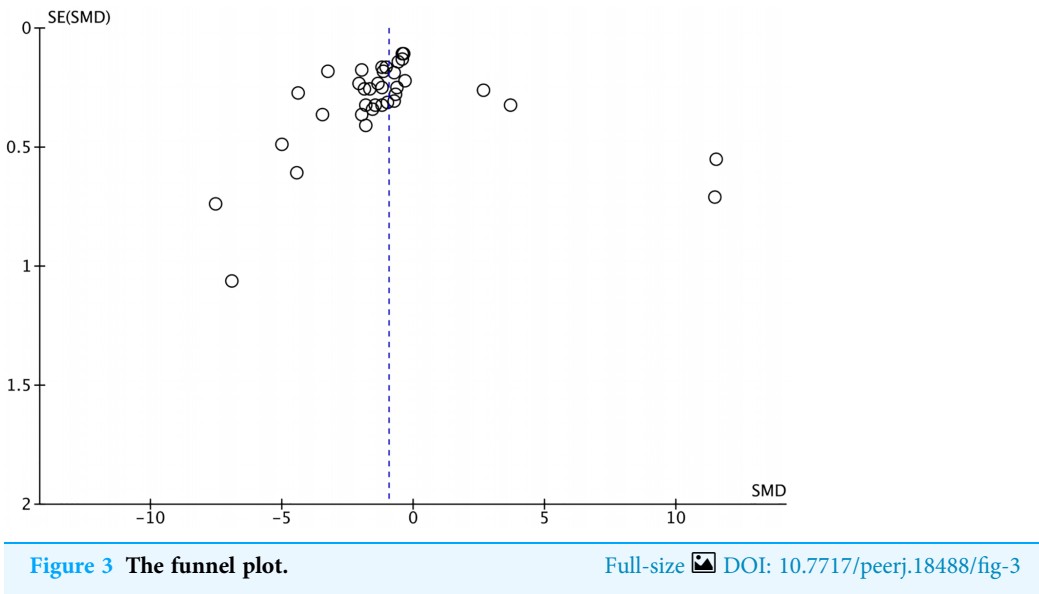

**Figure 3 The funnel plot.**

1.04 mmol/L. Based on the results obtained, it's evident that heterogeneity among subgroups had no statistical significance, indicating that TG ($P$ = 0.179), LDL ($P$ = 0.180), and HDL ($P$ = 0.162) levels did not originate from heterogeneous sources.

### Coronary artery lesion

*Stability of coronary atherosclerotic plaque.* The occurrence of CAD was closely associated with the formation and rupture of atherosclerotic plaques. We categorized the patients into three subgroups for further analysis: soft plaques, mixed plaques, and hard plaques. We found that between-subgroup heterogeneity had insignificant effects ($P$ = 0.075), implying that plaque stability didn't contribute to the observed heterogeneity.

*The number of diseased coronary vessels.* For subgroup analysis, we categorized the included studies based on the number of coronary artery lesions into three groups: single vessel lesion, double vessel lesions, and multi vessel lesions. Insignificant heterogeneity among subgroups was observed ($P$ = 0.152), suggesting that the observed heterogeneity cannot be attributed to the number of diseased coronary vessels.

### Classification of CAD

Patients were divided into three subgroups: SAP, UAP, and AMI. The findings indicated that no subgroups had significant effects ($P$ = 0.241), suggesting that the classification of CAD wasn't a factor of the observed heterogeneity.

### Publication bias

The funnel plot depicting the included articles was used to assess the publication bias (Fig. 3), and the results of Egger's tests (t = −0.30, $P$ = 0.769) and Begger's tests (z = −2.00, $P$ = 0.0483) were insignificant. No article was imputed after trim-and-fill correction, indicating that the results of the meta-analysis were robust and there was no publication bias.

## DISCUSSION

CTRP9, the closest paralog of adiponectin, plays a cardioprotective role in the CAD process due to its anti-inflammatory and anti-atherosclerosis features. Previous studies have investigated CTRP9 levels in patients with and without CAD, but their results were contradictory. Therefore, our research analyzed the circulating CTRP9 levels in CAD patients, and result showed that circulating CTRP9 levels in patients with CAD were significantly lower than those without CAD. A study has demonstrated that CTRP9 reduces cell apoptosis in myocardial tissue by activating the AMPK pathway, thereby reducing myocardial IRI and playing a protective role in the cardiovascular system (*Kambara et al., 2012*). Additionally, AMI inhibits both mRNA and protein levels of CTRP9 generation. CTRP9 supplementation ameliorates myocardial infarction cardiac remodeling through PKA-dependent pathway, improving survival rates of AMI mice (*Sun et al., 2013*). All these studies confirm that CTRP9 levels are lower in patients with CAD compared to those without CAD.

Due to the high heterogeneity, subgroup analysis and meta-regression were conducted to investigated the impact of different factors on CTRP9 levels. The results indicated that most characteristics of articles (year of publication, patients' age, and BMI) were not sources of heterogeneity, but the population distribution may contribute to the observed heterogeneity. As an adipocytokine, CTRP9 levels may be associated with obesity status. Studies have shown that obesity disrupts adipocytokine production and promotes the progression of diseases related to lipid metabolism disorders. However, there is no independent correlation between circulating CTRP9 levels and age, gender, BMI, TC, visceral fat, and *etc.* (*Hwang et al., 2014*). As for the population distribution, our results proposed the possibility of regional impact on serum CTRP9 of CAD patients. This impact may be due to ethnic and regional differences in genetic, environmental, and lifestyle. But it's worth noting that the majority of the included studies were conducted in China (92%, 23/25), which may introduce regional bias into our data. Clinical studies need to be conducted in more regions around the world.

Numerous studies have highlighted a significant correlation between diabetes and cardiovascular disease (CVD), which stands as the leading cause of morbidity and mortality associated with CVD (*Stamler et al., 1993*). Overexpression of CTRP9 has been shown to decrease insulin and blood glucose levels, indicating that CTRP9 may offer improvements in diabetes (*Wong et al., 2009*). Thus, diabetes or T2DM may be the cause of heterogeneity. However, results indicated that diabetes was not the source of heterogeneity, and there's no relation between diabetes or T2DM and serum CTRP9 levels. CTRP9 seems to play a role in the pathogenesis of diabetes (*Peterson et al., 2013*), but the problem becomes more complicated when vascular complications exist. Circulating CTRP9 levels decreases in patients with T2DM (*Song et al., 2023*), and its protective effect on atherosclerosis may be weakened.

Characterized as an adipocytokine, CTRP9 exhibits the highest expression within adipose tissue and plays a role in lipid metabolism (*Su et al., 2013*). The expression of CD36 protein in macrophages, which is a receptor of oxidized low density lipoprotein (ox-

LDL), can be diminished by CTRP9, indicating a protective effect of CTRP9 on the accumulation of ox-LDL (*Zeng et al., 2023*). These studies suggest a possible connection between blood lipids and CTRP9 level. Standard lipid blood tests include measurements of TG, LDL, and HDL. In our study, patients with CAD were stratified based on varying levels of these indicators, followed by subgroup analysis. The results reflected that TG, LDL, and HDL did not account for the observed heterogeneity. Some emerging evidence suggested that the low-density lipoprotein receptor-related protein 1 (LRP1)/calreticulin co-receptor system act on CTRP9-induced effects, and proprotein convertase subtilisin/kexin-9 (PCSK9) mediated reduction of LRP1 can diminish the effect of CTRP9 (*Potere et al., 2019*; *Rohrbach et al., 2021*). Nevertheless, the lipid-related signaling pathways by which CTRP9 protect against CVD are not yet comprehensive, so direct evidence is needed to clarify the mechanism (*Yang et al., 2016*).

CTRPs are widely expressed in various tissues, especially in mouse and human adipose tissue and plasma. Various studies have confirmed effect of excess adipose tissue on increasing coronary atherosclerosis. Recently, *Asada et al. (2016)* also found that plasma CTRP9 levels were associated with atherosclerosis in patients with type 2 diabetes without renal dysfunction. CTRP9 can enhance atherosclerotic plaque stability by attenuating vascular smooth muscle cells (VCMCs) proliferation and neointimal formation (*Uemura et al., 2013*), increasing the expression of adiponectin receptor (AdipoR) 1 (*Li et al., 2015*) and reducing the level of pro-inflammatory cytokines (*Zhang et al., 2016*). However, the results suggested that plaque stability and the number of coronary artery lesions were not heterogeneous sources. More research is needed in the future.

CAD can be clinically classified into three categories: UAP, SAP, and AMI. CAD classification was used as subgroups for analysis, and the between-subgroup heterogeneity manifested no significant change. There is no worldwide consensus on whether CAD should be classified or how many categories it should be divided into. Additionally, the included articles did not provide an explanation for the criteria used in classifying CAD. Therefore, classifying CAD into subgroups for analysis may lack accuracy. More mechanistic research is needed to elucidate the relationship between CAD classification and CTRP9 levels.

In addition to the above factors, there may be other factors affecting the observed heterogeneity. *Li et al. (2020)* reported that patients with CAD who experienced moderate to severe obstructive sleep apnea (OSA) exhibited lower CTRP9 levels compared to those without OSA or with mild OSA. *Fadaei et al. (2023)* discovered a increase in CTRP9 levels among OSA patients. Besides, studies showed a negative correlation between CTRP9 levels and the severity of peripheral arterial disease in T2DM (*Jiang et al., 2018*). Studies demonstrated a positive correlation between adiponectin and CTRP9 (*Wang et al., 2015*). In addition, evidence showed that serum CTRP9 may also be correlated to adiponectin, systolic pressure, CRP, leptin, and *etc.* (*Bai et al., 2017*; *Hwang et al., 2014*; *Liu et al., 2014*). Regretfully, data from the included studies were insufficient to conduct further subgroup analysis, and more factors affecting heterogeneity need to be explored in the future.

Our research still has the following limitations. First, all studies included in this meta-analysis were retrospective, making it challenging to establish a causal relationship between CTRP9 levels and CAD. Second, two articles we incorporated presented non-normally distributed data, necessitating the use of median and quartiles for data description (*Hu & Lin, 2021*; *Liu et al., 2022b*). Nonetheless, we proceeded with conversions to describe CTRP9 levels using mean and standard deviation, which could introduce errors. Third, SMDs were pooled in this meta-analysis due to their uniform unit across studies. Fourth, most included studies were conducted in China (92%, 23/25), which may introduce regional bias. Besides, it is difficult to draw rigorous conclusions with only 25 articles reviewed, and more accurate studies with richer data may be conducted in the future.

## CONCLUSIONS

In conclusion, our meta-analysis has substantiated the significant reduction of circulating CTRP9 levels in patients with CAD compared to those without CAD, and the association may be modified by population distribution. Our findings align with existing fundamental research on the association between CTRP9 and CAD. However, given that all studies included in this meta-analysis were observational in nature, larger population-based studies and studies in more regions are warranted to further strengthen the credibility of this conclusion.

### Funding
The authors received no funding for this work.

### Competing Interests
The authors declare that they have no competing interests.

### Author Contributions
- Ziyi Zhu conceived and designed the experiments, performed the experiments, analyzed the data, prepared figures and/or tables, authored or reviewed drafts of the article, and approved the final draft.
- Qingsheng Niu conceived and designed the experiments, performed the experiments, analyzed the data, authored or reviewed drafts of the article, and approved the final draft.
- Shiyuan Tang conceived and designed the experiments, performed the experiments, analyzed the data, authored or reviewed drafts of the article, and approved the final draft.
- Yaowen Jiang conceived and designed the experiments, authored or reviewed drafts of the article, and approved the final draft.

### Data Availability
  This is a systematic review/meta-analysis.

## Supplemental Information

Supplemental information for this article can be found online at http://dx.doi.org/10.7717/peerj.18488#supplemental-information.

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
