# Peer review of "Association between circulating CTRP9 levels and coronary artery disease: a systematic review and meta-analysis"

_PeerJ, doi:10.7717/peerj.18488_

## Round 0.1 · original submission · Major Revisions

Please address the critiques of all 3 reviewers and amend your manuscript accordingly.

Reviewer 1 ·

Basic reporting

Writing needs to be improved. In most places, the symbol (,) is missing,
Literature and background information are provided.
Self-contained with relevant results to the hypothesis

Experimental design

Research question well defined and meaningful
The study has provided insights into the circulating CTRP9 and CAD
Methods described with sufficient details

Validity of the findings

Underlying data and Conclusion are well stated

Additional comments

1. Why is CTRP-9 specifically chosen among the adiponectin paralogs? Different CTRPs are found to be associated with CAD, and the reason behind selecting CTRP-9
2. The heterogeneity and association of CTRP9 with Obesity/sleep apnea and vascular complications add value to the manuscript.
3. Adiponectin is similar to CTRP-9. It will be good to know its association with CTRP-9 in CAD
4. CTRP-9 is elevated in hypertension-related atherogenesis. It must be added as a heterogeneity factor since it might impact the study.

·

Basic reporting

The manuscript is written in clear and professional English, but the subject matter is overly broad, which detracts from the clarity and focus of the research. The literature review is limited, with only 25 papers referenced, which is insufficient to support a comprehensive meta-analysis. As a result, the background and context provided are not robust enough to substantiate the study's objectives. Additionally, the conclusions drawn seem to align with common sense and do not offer new insights or contributions to the field.

Experimental design

The research question, while relevant, is not sufficiently narrowed down to provide meaningful insights. The scope is too broad, making it difficult to address the specific relationship between circulating CTRP9 levels and coronary artery disease (CAD) effectively. The investigation lacks depth, particularly in terms of the number of studies included for meta-analysis. With only 25 papers reviewed, it is challenging to perform a rigorous meta-analysis that can lead to valid conclusions. The methods are described adequately, but the study's design does not meet the standards for a high-quality systematic review or meta-analysis.

Validity of the findings

The findings of the study appear to be based on common knowledge, with no novel contributions or significant impact on the field. The limited number of studies included in the analysis undermines the robustness of the results. Moreover, the final conclusions seem to reflect commonly accepted knowledge rather than offering new perspectives or breakthroughs. As a result, the study's findings do not provide meaningful advancements in understanding the relationship between CTRP9 levels and CAD.

Additional comments

Two suggestion:
1) Expand the Scope and References: Consider broadening the focus to include a wider range of related topics, and incorporate a more extensive set of references to support a comprehensive review. This will help in providing a well-rounded perspective and strengthen the foundation of the analysis.

2) Offer New Insights and Recommendations: Aim to develop novel insights or actionable recommendations that can contribute to current and future research in the field. Providing fresh perspectives or suggesting directions for future work will enhance the manuscript's relevance and impact on the scientific community.

Reviewer 3 ·

Basic reporting

NA

Experimental design

NA

Validity of the findings

NA

Additional comments

The paper presents a comprehensive analysis aimed at elucidating the relationship between CTRP9 levels and coronary artery disease (CAD). Main Findings: Patients with CAD showed significantly lower circulating levels of CTRP9 compared to controls, suggesting that lower CTRP9 levels are associated with CAD. And a funnel plot indicated potential publication bias.

However there are few issues the authors need to address:
1. The authors report high heterogeneity (I² = 98%) which they partially attributed to differences in the stability of coronary plaques and types of CAD. However, the persistence of high heterogeneity even after subgroup analysis suggests there might be other unexplored sources of variability. This raises concerns about the generalizability of the findings.

2. The majority of studies included are from China (92%).This limit the generalizability of the results to global populations due to ethnic and regional differences in genetic, environmental, and lifestyle factors that influence CTRP9 levels and CAD.

3. The authors converted non-normally distributed data into mean and standard deviations. This approach, while making it easier to compute standardized mean differences, can introduce errors and is not always methodologically sound.

4.The funnel plot indicated potential publication bias. Authors should consider using more robust methods to assess and correct for publication bias, such as trim and fill analyses.

---

## Round 0.2 · accepted · Accept

All issues pointed by the reviewers were addressed and revised manuscript is acceptable now.

Reviewer 3 ·

Basic reporting

NA

Experimental design

NA

Validity of the findings

NA

Additional comments

The authors have addressed all of my concerns.